# Inborn Errors in the LRR Domain of Nod2 and Their Potential Consequences on the Function of the Receptor

**DOI:** 10.3390/cells10082031

**Published:** 2021-08-09

**Authors:** Shamila D. Alipoor, Mehdi Mirsaeidi

**Affiliations:** 1Department of Molecular Medicine, Institute of Medical Biotechnology, National Institute of Genetic Engineering and Biotechnology, Tehran 1497716316, Iran; shamilaalipoor@gmail.com; 2Division of Pulmonary and Critical Care, College of Medicine-Jacksonville, University of Florida, Jacksonville, FL 32209, USA

**Keywords:** NOD2, NLRs, ER stress, autophagy, innate immunity

## Abstract

The innate immune system plays a critical role in the early detection of pathogens, primarily by relying on pattern-recognition receptor (PRR) signaling molecules. Nucleotide-binding oligomerization domain 2 (NOD2) is a cytoplasmic receptor that recognizes invading molecules and danger signals inside the cells. Recent studies highlight the importance of NOD2′s function in maintaining the homeostasis of human body microbiota and innate immune responses, including induction of proinflammatory cytokines, regulation of autophagy, modulation of endoplasmic reticulum (ER) stress, etc. In addition, there is extensive cross-talk between NOD2 and the Toll-like receptors that are so important in the induction and tuning of adaptive immunity. Polymorphisms of NOD2′s encoding gene are associated with several pathological conditions, highlighting NOD2′s functional importance. In this study, we summarize NOD2′s role in cellular signaling pathways and take a look at the possible consequences of common NOD2 polymorphisms on the structure and function of this receptor.

## 1. Introduction

The innate immune system provides the first line of defense against danger and relies primarily on pathogen recognition receptors (PRRs) to do so [1]. PRRs are the immune system players that recognize the molecules frequently found in pathogens or released by damaged cells (respectively known as pathogen-/damage-associated molecular patterns (PAMPs or DAMPs)) [2].

PRRs are categorized into four distinct functional groups: (1) Toll-like receptors (TLRs), (2) retinoic acid-inducible gene (RIG)-I-like receptors (RLRs), (3) C-type lectin receptors (CLRs), and (4) nucleotide-binding oligomerization domain-like receptors (NLR) [3]. 

NLRs are intracellular immune receptors conserved in both animals and plants. While some NLR proteins are involved in early embryogenesis and regulate the expression of major histocompatibility complex (MHC) molecules, certain NLR proteins play critical roles in recognizing damage-associated molecular patterns and in triggering immune responses [4]. 

The major PPRs, including TLRs, detect and capture pathogens on the cell surface or within endosomes, while NLRs are cytoplasmic receptors and detect their ligands in the cytosol, thereby providing another level of cell protection [5]. 

Nucleotide Binding Oligomerization Domain Containing 2 (NOD2) is a well-known member of the NLR family, which is expressed primarily in immune and epithelial cells [6,7]. This receptor detects a fragment of bacterial cell wall known as peptidoglycan muramyl dipeptide (MDP) and subsequently activates the signaling pathways, leading to proinflammatory cytokine production [8]. It has been shown that the polymorphisms in the NOD2 gene contribute to failure in microbial detection and are associated with increased susceptibility to some infectious diseases and granulomatous inflammation [9]. 

In this study, we first summarized the recent findings on the role of NOD2 in cell vital pathways and then assessed how the polymorphisms in NOD2 gene could affect the structure and function of the receptor using computational analysis.

## 2. NLR Family and Structure

NOD-like receptors (NLRs) are evolutionally conserved proteins, belonging to the PRR family [5]. NLRs are also considered a large family of cytoplasmic receptors consisting of 22 members in humans and 34 members in mice [10]. They have an important role in the triggering and development of innate immune responses thorough sensing intracellular danger signals [11]. 

NLR proteins share a conserved triple domain structure containing a C-terminal leucine-rich repeat (LRR) domain, a central nucleotide-binding and oligomerization domain (NOD/NBD) (also known as NACHT domain), and a N-terminal protein–protein interaction domain (Figure 1) [7]. The C-terminal LRR domain is responsible for the detection of PAMPs and DAMPs and negatively regulates protein activity. The central NOD domain has ATPase and nucleotide binding activity, which is critical in protein oligomerization and function. The NOD domain contains a proximal helical domain 1 (HD1), a distal helical domain 2 (HD1), and a winged helical domain (WHD) [7] (Figure 1). The N-terminal effector domain is responsible for interacting with the downstream signaling molecules. 

Based on the type of effector domains, the NLR family is divided into several subfamilies including NLRA containing an acidic transactivation domain (AD), NLRB (also known as NAIP) with a Baculovirus IAP Repeat (BIR) domain, NLRC with a caspase activation and recruitment domain (CARD), and NLRP with PYRIN domains (PYD) [12]. 

The NLRA and B subfamilies are involved in antiapoptotic functions and the transcription activation of MHCII via their intrinsic acetyl transferase (AT) activity. NLRC is one of the largest subfamilies of NLRs, consisting of six members (NOD1-5 and class II trans activators) that are characterized by their CARD effector domains [13]. 

The effector CARD domains have an important role in NLR’s downstream functions and interact with other CARD-containing proteins through homophilic interactions. NOD2 contains two tandem CARD effector domains and can interact with a wide variety of proteins containing the CARD domain (Figure 1) [12]. Here, we provide a brief review of NOD2 mechanisms and functions.

**Figure 1 cells-10-02031-f001:**
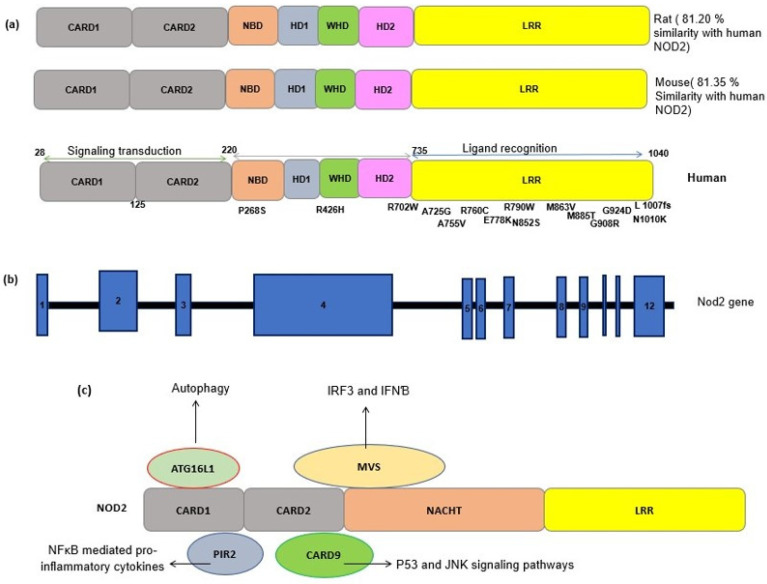
(**a**) Schematic representation of the NOD2 protein structure in a human, a mouse and a rat. The sequence identity of NOD2 gene in the rat and mouse with the human NOD2 gene is estimated by pairwise BLAST [14]. Common SNPs have been shown alongside the protein domains. CARD, caspase recruitment domain; NACHT, nucleoside triphosphates’ (NTPase) domain; LRR, leucine-rich repeats. (**b**) A schematic representation of the NOD2 gene. NOD2 is composed of 12 exons (blue rectangles). The numbers inside the blue rectangles indicate the exon numbers. (**c**) A schematic of the interactions between NOD2 and other cellular proteins. The interaction of activated NOD2 with PIR2 activates the NF-kB pathway. The NOD2 interaction with Autophagy-related 16-like 1 (ATG16L1) induces autophagy machinery assembling. NOD2 interacts with the adapter protein mitochondrial antiviral signaling protein (MAVS) upon sensing ssRNA, active interferon-regulatory factor 3 (IRF3) and consequently production of interferon β (IFNβ).

## 3. NOD2: Cellular and Molecular Mechanisms 

NOD2 acts through the mitogen-activated protein kinase (MAPK), inflammasome-associated, and NF-κB pathways which are considered the three main cell signal transduction pathways [15]. In the epithelial cells, NOD2 molecules are committed to the synthesis of anti-pathogenic peptides [16]. The expression level of specific antimicrobial α-defensins was significantly decreased in the Paneth cells of NOD2 knockdown mice [17]. Additionally, NOD2 can recruit Autophagy-related 16-like 1 (ATG16L1) and subsequently induce autophagy after activation (Figure 1) [18].

NOD2 is primarily activated upon sensing a component of bacterial peptidoglycan named N-acetyl muramyl dipeptide (MDP). It has been shown that NOD2 interacts with a wide variety of proteins. Mycobacterial N-glycolyl muramyl dipeptide and viral ssRNA are also the ligands of NOD2, which can active their associated signaling pathways [2]. 

Until the ligand activation, NOD2 is maintained in an inactive, autoinhibited conformation in the cell through interactions of the NOD domain with LRR domains and cellular chaperones, such as heat shock protein 90 (HSP90). 

Upon activation, the C-terminal LRR domain of NOD2 undergoes a conformational change and exposes the CARD domain, which allows it to interact and oligomerize with the CARD domain in the adaptor molecule RIP2 (receptor-interacting protein 2) through a homophilic interaction. Upon oligomerization, activated PIR2 applies lysine 63 (K63)-linked polyubiquitination at lysine 209 of the kinase domain. This ubiquitination promotes recruitment of TAK1 and NEMO (the NF-kB essential modulators). The activation of TAK1 and NEMO promotes phosphorylation of the IKKβ kinase, which is a key kinase in the NF-κB signaling pathway. Phosphorylated IKKβ degrades IκB and subsequently actives NF-κB family transcription factors and the production of inflammatory chemokines [19] (Figure 2). 

Conversely, the activation of NOD2 by viral ssRNA leads to the production of interferon β (IFNβ) through an alternative pathway by recruiting an adapter protein, a mitochondrial antiviral signaling protein (MAVS), and an activating interferon-regulatory factor 3 (IRF3) [20] (Figure 1). 

NOD2 also may trigger an autophagic pathway upon the detection of bacterial MDP by recruiting ATG16L1 to the bacterial entry site, which results in the engulfment of invading bacteria by autophagosomes formation [21]. The NOD2 signaling pathways are summarized in Figure 2.

### 3.1. NOD2 in Immune Pathways

NOD2 drives the innate and adaptive immune response against pathogen and danger signals through different mechanisms, including triggering of immune signaling pathways such as NF-κB and MAPK pathways [2]. These processes promote recruitment of neutrophils and immune cells to the site of infection and lead to an enhanced expression of proinflammatory factors and antimicrobial agents [22,23,24]. NOD2 actives the Notch1 signaling pathway in macrophages by mediating the Notch1–PI3K axis, which results in macrophage survival and regulates the expression of anti-inflammatory genes, including IL-10 [25]. Along with its essential function in innate immunity, NOD2 plays a critical role in the adaptive immune system. It has been shown that MDP-induced NOD2 activation promotes the development of Th17 cells from memory Th cells [16] and increases the level of IL-17 and IL-22 [26,27,28]. 

Upon activation, NOD2 changes the phenotype of APCs and other immune cells by increasing the expression level of surface co-stimulatory molecules [29]. Th2 immunity is induced upon NOD2 activation and upregulation of the OX40 ligand on the dendritic cells, which results in the production of IL-4 and IL-5 [30]. 

Additionally, activated NOD2 educates the dendritic cells to induce the expression of interleukin-17 by memory T cells [28]. The co-stimulation of NOD2 and TLRs synergistically induces L-12, IFN-γ, or IL-6 production and activates Th1-associated immune responses [26,31]. The underlined mechanisms in this process are not clearly understood, but it is suggested that TLR stimulation promotes the internalization of MDP, which makes it available to be captured by NOD2 [32]. 

NOD2 regulates the expression of immune regulatory miRNAs, including the miR-29 family. miR-29 suppresses IL-12p40/IL-23 expression and Th17 CD4^+^ T cell responses [33]. It has been reported that, in the dendritic cells (DCs) from patients with mutant NOD2 variants, the induction of miR-29 following NOD2 activation is impaired [33].

NOD2 also triggers antiviral immune responses upon sensing of virus-associated PAMPs, such as viral genomes in the infected cells [34]. Activated PIR2, the mediator of NOD2 downstream signaling pathway, binds TNF receptor-associated factor3 (TRAF3); activates the TANK binding kinase 1(TBK1) and IKKε, which subsequently phosphorylate and active interferon response factor 3 (IRF3); and triggers IFN-stimulated response elements (ISGs) [2]. 

NOD2 also activates IRF3 through the recruitment of mitochondrial antiviral signaling proteins (MAVS), which result in the induction of type I IFNs and antiviral immune responses [34].

Furthermore, NOD2 can bind to 2′-5′-oligoadenylate synthetase type 2 (OAS2), a dsRNA binding protein, and can enhance the function of RNase-L to control viral infection [35].

NOD2 has a determinant role in BCG-induced innate memory through epigenetic reprogramming in macrophages, and it has been recently suggested that NOD2 signaling could play a critical role in tuning innate immune responses against SARS-CoV2 in BCG-vaccinated individuals [36]. Interestingly, the reduction in NOD2 expression in macrophages was reported in MERS-CoV infection as one of the viral strategies to circumvent innate immunity [37].

### 3.2. NOD2 and ER Stress

The endoplasmic reticulum (ER) is an important organelle with critical roles in the regulation of several functions in eukaryotic cells, including synthesis, folding, and traffic of proteins. The accumulation of mis-folded or unfolded proteins induces ER stress and triggers the unfolded protein response (UPR) to limit cellular damage. ER stress plays a critical role in inflammatory reactions, and dysregulated UPR is associated with several immune-mediated diseases [38].

There are generally three main UPR signal activator proteins, including activating transcription factor 6α/β (ATF6), PKR-like ER kinase (PERK), and inositol requiring enzyme 1α/β (IRE1α3) [39]. Upon UPR activation, the cytoplasmic region of IRE1α triggers an inflammation cascade by recruiting TRAF2 to the ER membrane and by subsequently triggering the c-Jun N-terminal kinase (JNK) and NF-κB pathways.

It is not completely clear which PRRs are involved in UPR, although recent evidence has suggested NOD1 and NOD2. The mechanism of NOD activation by ER stress is also still unclear. However, it is suggested that they act through ER stressors, such as thapsigargin and dithiothreitol [40], and increase the expression level of pro-inflammatory cytokines, such as interleukin-6 (IL-6) [40].

Furthermore, the pro-inflammatory reactions induced by injection of ER stress inducers were suppressed in NOD1/2 knockout mice [40]. These findings indicate the role of NOD2 in promoting ER stress-induced inflammatory responses.

A better understanding of how ER stress and NOD2 are linked will clarify the role of ER stress in host defense. ER stress may have a determinant role in the pathogenesis of NOD2-associated inflammatory diseases and in increasing individual susceptibility to infection.

The contribution of NOD2 to UPR responses indicates the link between ER stress and innate immunity and provides a new avenue for therapeutic methods for treating inflammatory and infectious diseases [41]. 

### 3.3. NOD2 and Autophagy

Autophagy is a self-regeneration process of cell protein and organelle turnover that cells undergo to maintain their physiological balance. Autophagy is initiated in response to starvation, intracellular infections, formation, and accumulation of protein aggregate and to oxidative stress and other conditions wherein a cell decides to remove its damaging cytoplasmic components [42]. Autophagy has an important role in maintaining cell homeostasis and in regulating immune response [42]. 

NOD2 contributes to the autophagy process by recruiting the autophagic protein ATG16L1 to the site of bacterial entry and subsequently triggers autophagic machinery to engulf the invading pathogen [43]. 

Interestingly, the autophagy process was suppressed in mice with NOD2 deficiency [44]. Additionally, macrophages with deficiency in the TLR2 and NOD/RIP2 pathways failed to trigger autophagy upon bacterial infection. Furthermore, ATG16L1 knockdown mice increased their level of cellular NOD2, suggesting that this critical autophagic protein can be involved in the regulation of NOD2 [45].

In some variants of Crohn’s disease, NOD2 lost this ability to induce autophagy upon MDP sensing and removing the intracellular bacteria [46]. 

NOD2-dependent autophagy also is crucial in efficient antigen presentation by DC and thus consequent stimulation of the CD4^+^ T cells against bacterial antigens [47].

### 3.4. NOD2 and Its Importance in Pulmonary Diseases

The respiratory system is exposed to a high volume airflow containing a large number of inhaled damaged particles. The innate immune system plays a key role in various infectious and non-infectious disorders of the lung by sensing damage and infections [48]. NOD2 has a crucial role in the pathogenesis of several pulmonary diseases, including mycobacterial pulmonary diseases, COPD, asthma, and pulmonary inflammatory diseases. The receptor plays an essential role in the diagnosis of lung microbial infections, such as mycobacterial infection, and regulates the host response to M. tuberculosis (M.tb) in the lung [49,50]. Polymorphism in the NOD2 gene is strongly associated with susceptibility to mycobacteria infections. It has been shown that missense E778K and G908R are associated with recurrent pulmonary nontuberculous mycobacterial infections [51].

Furthermore, a high expression of NOD2 is reported in BAL cells from patients with sarcoidosis and Behcet’s disease (BD) and with pulmonary presentations that may be responsible for lung inflammation [52]. Impaired NOD2 function and the resultant aberrant inflammation could promote the development, progression, and exacerbation of COPD [53].

NOD2 polymorphisms have been associated with increased risk of developing asthma [54,55,56]. It was reported that the NOD2 gene is significantly overexpressed in human airway smooth muscle cells (HASMC) in asthma patients and could be considered a potential diagnostic biomarker and a therapeutic option in this disease [57,58]. 

NOD2 deficiency also leads to the exacerbation of hypoxia-induced pulmonary hypertension and promotes pulmonary vascular smooth muscle cell (PASMC) proliferation and vascular remodeling [59].

NOD2 mutations plays a major role in the pathogenesis of noninfectious granulomatous diseases, including sarcoidosis and Crohn’s disease (CD), which might be accompanied by pulmonary manifestations and lung damage. 

In sarcoidosis, lung involvement mostly presents with bilateral hilar adenopathy manifestation [60]. Polymorphisms in the NOD2 gene are strongly associated with an increased susceptibility for developing sarcoidosis [61,62,63]. The G908R NOD2 variant was reported in a familial case of sarcoidosis [64]. Furthermore, combinations of polymorphisms in NOD2 and immune-related genes have shown a significant association with the development of this disease [64].

The NOD2 gene is also known as the major genetic risk factor for Crohn’s disease (CD) and Inflammatory bowel disease (IBD) [2]. CD is a systemic illness affecting various organs. Lung manifestation of CD is relatively rare [65]. However, CD may engage small and large airways and may lead to lung parenchymal diseases and pulmonary embolism. Bronchiectasis is the most commonly reported form of respiratory disease in CD [66,67].

In addition, increased frequency of asthma has also been reported in IBD patients. IBD patients with asthma have a more severe respiratory course and more significant reduction in lung function compared with asthmatics without IBD and have shown an increased risk of fatal asthma [68]. 

## 4. NOD2 Genetics and Polymorphism

The human gene encoding for the NOD2 receptor is CARD15, located on chromosome 16q12.1. The NOD2 protein has 104 amino acids with a molecular weight of 110 kDa, which is a multifunctional receptor. As NOD2 has many important roles, mutations in its gene may have serious consequences in vital cellular functions and immunity. NOD2 is a repository of genetic variants, most of which are associated with pathological conditions. Many previous studies have reported the association of NOD2 polymorphisms with inflammatory diseases (Table 1) [23]. As LRR is the ligand binding domain of the NOD2 receptor, mutations in this region may affect either responses to MDP or the downstream pathways [69]. The nonsense mutations in this region also may abolish the conformational changes needed for MDP binding and receptor activation and thus may lead to receptor loss-of-function. 

As the residues within the LRR domain have a critical role for the response to MDP, we predicted the potential effects of common LRR polymorphisms on the structure of NOD2 by computational analysis and discuss how these variations could change NOD2 function in interaction with pathogenic particles.

### 4.1. Characterization and Functionality of Variants

Based on the conservation analysis of the missense variants in the LRR domain of NOD2, the A725G, A755V, E778K, N852S, V793M, and G908R are in highly conserved regions; thus, substitutions in these positions would be high risk and likely to damage receptor function. However, R790W, E843K, A885T, and G924D are located in the variable region of the protein, which indicates that mutations in this region can be accepted and would probably not be damaging for the protein structure or function (Figure 3).

Furthermore, the evaluation of the functionality of these missense variants has shown that seven SNPs (A755V, E778K, R790W, V793M, N852S, A885T, and G908R) are critical for NOD2 function and thus could be damaging for the receptor function upon mutation (Table 2).

### 4.2. The Effect of Mutations on the NOD2 Structure and Interatomic Interactions

The prediction of the effects of these SNPs on the structure and interatomic interactions in the mutant receptors indicated that these variations could disrupt the interatomic interactions, including the H-bond and ionic interactions, and the salt bridge and H-bond with the adjacent molecules (Appendix A).

Predictions based on the size, charge, and hydrophobicity properties of the substituted residues in comparison with the wild type residues have shown that variants A755V, E778K, N852S, G908R, and G924D can induce damage to the protein structure while other mutations could be tolerated by the protein without affecting its structure (Appendix A). 

### 4.3. Determination of the SNPs’ Effects on Protein Stability and Flexibility

Measuring the similarities between the native and mutant structural models based on RMSD and TM-score indicated that the structures demonstrated almost the same fold.

However, the effects of SNPs on the stability of the NOD2 structure based on the reliability index (RI) and the free energy change values by measuring Delta Delta G (DDG) indicated that, except A885T, all of the substitutions decreased the total structural stability of the mutant receptor (Appendix A).

These variants also affect the structural flexibility in the mutated proteins. 

Predictions based on vibrational entropy indicated an increase in the rigidity of LRR domains in G908R, A725G, N852S, and M863V mutant structures while it demonstrated that A755V, R760C, R790W, E843K, A885T, and G924D substitutions lead to increased flexibility in the LRR domain of the receptor (Appendix A). These mutations did not affect the flexibility in the other domains of the mutant proteins. However, in the case of E778K, mutations from E to K increased the total molecular flexibility and increased protein flexibility in the LRR domain, while it increased rigidity in the HD1, HD2, and WHD domains of the NOD2 structure (Appendix A).

In addition, these mutations could significantly change the interaction between residues in the mutant proteins and significantly change the interaction of new residues with the adjacent residues in the mutant proteins, which would affect proper folding of the mutant molecules (Appendix A). The biggest effect was observed upon G908R substitution (Appendix A), which led to changing the position and conformation ligand binding site in this molecule [88].

### 4.4. Prediction of Posttranslational Modification Sites

The prediction of posttranslational modification sites in the mutant protein indicated that, in N852S and A885T, the new S and T residues provide new phosphorylation or methylation sites.

Moreover, the new residues in E778K and E843k substitutions introduces a new ubiquitination site in the mutated proteins.

In addition, the new residues in positions 908 and 778 introduce new peptide cleavage sites in the mutant proteins that were not present in the wild type (Appendix A). Introducing a new ubiquitination site in the protein may impress downstream interactions, and introducing new peptide cleavage sites upon mutation makes the mutant protein susceptible to proteases and reduces its half-life and stability. 

### 4.5. Protein–Protein Interaction (PPI) Analysis

To determine the consequence of the SNPs on the NOD2-related downstream molecules and pathways, we assessed its interactions in a PPI network. We first constructed a network of the NOD2-related proteins based on their direct PPI neighbors and their interactions, and then, the highly connected region from this network and its related pathways were extracted. The results identified receptor-interacting serine/threonine-protein kinase 2 (RIPK2), autophagy related 16 like 1 (ATG16L1), caspase recruitment domain-containing protein 9 (CARD9), tumor necrosis factor receptor-associated factor 6 (TRAF6), and inhibitor of Nuclear Factor Kappa B Kinase Regulatory Subunit Gamma (IKBKG) as the highly connected molecules with NOD2 that are primarily involved in apoptosis and immune-related pathways (Appendix A).

These computational predictions of the common SNPs in the LRR domain of the NOD2 showed that these mutations could impress the ligand recognition, receptor stability, LRR conformation, and interaction with other proteins. 

Furthermore, the changed flexibility in the region of mutations could affect the transformation from the inactive to active form of the protein and could change the susceptibility to the ligand, resulting in receptor loss-of-function.

Based on the predictions, it may be suggested that G908R, E778K, N852S, and A755V substitutions were most likely to be harmful and induced the most damaging effects on the structure of the receptor.

Tanabe et al. conducted a mutational analysis of the NOD2 protein and reported that the G908R mutation causes loss of function (LOF) [89]. However, they could not find the underlying mechanism for LOF. The current study shows that G908R decreases the flexibility in the WHD–HD2–HD1 domain and thus slows conversion to the active form of the receptor while increasing the rigidity in this region. Additionally, in the position of G908, the torsion angles in this region are unusual, and glycine is flexible enough to compensate. The presence of a larger R-side chain in this region may induce an incorrect conformation to the local backbone and disturbs the local structure.

Furthermore, upon introduction of the G908R mutation in the LRR domain, the pattern of contacts between the residues and interatomic interactions significantly changes. We found that these dramatic changes contribute to changing the position and conformation of the ligand binding site. This may play an essential role in the MDP non-responsiveness of the mutant protein that was previously reported by in vitro experiments [51,90].

It has been shown that, in the LRR–HD1 interface, the α3 helix (HD1) is packed against the α1 and α2 helices of the LRR via different interatomic contacts [90]. E778 is located in α2 helices, and its interaction with its adjacent residues plays an important role in this interface. Our data shows that the substitution of E to K in position 778 significantly changed the interatomic interactions in this region. The charge difference in the substituted residue also induces a repulsion with other residues in the protein or ligands [90]. We observed that this mutation decreased the total flexibility of the protein, making it less capable of structurally converting to the active form.

In a recent study, it was reported that missense E778K and G908R are associated with recurrent pulmonary nontuberculous mycobacterial infections [51]. Our study revealed that these substitutions change the flexibility of the critical region of the receptor structure needed for transformation of the inactive to active form of the protein and thus to change susceptibility to the ligand. This may be responsible for receptor LOF and increased susceptibility to the pulmonary nontuberculous mycobacterial infections.

A755V is located on an α helix in the LRR domain. The mutant residue(V) is bigger than the wild type and is less compatible with α-helices as a secondary structure in this region, which may disturb correct folding. Our result has shown that this variant leads to a change in flexibility in this position and could affect the receptor responding to MDP. It has been reported that this mutation is associated with an immunologic significance including IBD1 and ulcerative colitis [91]. In fact, our study shows that this substitution changes the structure of NOD2 in the LRR domain via changing the interatomic interactions, total flexibility, and stability of the receptor.

Our analysis shows that the N852S mutation could be very deleterious to the structure and function of a receptor due to significant induced structural changes. We observed that, in the N852S variant, the interatomic reactions in the mutant structure decreased, which can be explained by the smaller size of the mutant residue. On the other hand, the mutant residue is more hydrophobic, which leads to a loss of hydrogen bonds and disturbs correct folding in the mutant region. Furthermore, this substitution leads to increased rigidity in the LRR domain, which could affect the transformation changes required for ligand recognition. 

Beside the presence of SNPs in the NOD2 structure, which may predispose the body to infection, environmental factors and other transcriptional factors, including miRNAs, might play a role in poor outcomes. The presence of other mutations in the downstream molecules that may boost or compensate for the effects of these inborne errors is another possibility for the increased susceptibility to infections.

This study helps to provide a comprehensive view of the potential role of inborne errors in the LRR domain of NOD2 on the molecular mechanisms of diseases, errors which may facilitate the development of the associated immune diseases.

## 5. NOD2: Recent Progress and Future Research Perspectives

Current findings on NOD2 indicated that this molecule plays a critical role in the innate immunity and regulation of inflammatory pathways. 

The stimulation of NOD2 by MDP triggers several immune pathways so this molecule has been considered a potential therapeutic target in immune diseases. One of the suitable therapeutic strategies could be through its downregulation.

Direct NOD2 inhibitors have not been introduced yet because of the technical challenges of targeting oligomeric proteins. However, RIPK2, which is a key molecule in NOD2 downstream signaling pathways, is interesting as a suitable target for treatment strategies [92,93]

The RIPK2 inhibitors inhibited the murine experimental colitis through the suppression of the polyubiquitination and kinase activity of RIPK2 [94,95]. Another study suggested that the ATP-binding domain in RIPK2 also could be targeted to interfere with the RIPK2–XIAP interaction and the modulation of NOD signaling [96]. 

Many recent experimental and clinical studies have focused on the applicability of RIPK2 inhibitors as a new therapeutic target in NOD2-dependent autoimmune diseases including IBD [92,97].

Targeting of the NOD2 immune pathways also have been investigated as vaccine adjuvants. Using TLR4 ligand (monophosphoryl lipid A (MPLA)) as a vaccine adjuvant was approved for the first time in 2017 to stimulate immunity against influenza antigens [98]. However, using PRR agonists such as NLRP3 [99], TLR3 [100], TLR7 [101], or TLR9 [102] ligands as vaccine adjuvants are currently being examined in clinical trials [103].

NOD2 agonists are investigated as efficient mucosal adjuvants. Recently, a new chimeric TLR7/NOD2 ligand was introduced as a potent adjuvant to stimulate both systemic and mucosal immune responses through an optimal DC maturation process [103]. 

In addition, MIS416, a nontoxic microparticle, is introduced as a cancer vaccine adjutant in a first-in-human clinical trial and activates the immune functions via NOD2- and TLR9-related pathways [104]. 

It is suggested that NOD2 may play an important role in the outcome of kidney cancer and that the expression level of NOD2 gene is associated with the prognosis of kidney cancer and has potential as a biomarker for the survival of patients with kidney cancer.

NOD2 may play a role in carcinogenesis via regulation apoptosis and inflammation. NOD2 variants have been reported to be associated with risk of cancers in European Americans [105,106,107]. Therefore, targeting of the molecules involved in NOD2 signaling pathways could be a potential target of cancer therapy.

Interestingly, it has recently been suggested that BCG-induced trained immunity via NOD2 may play a key role in the control of SARS-CoV2 infection. SARS-CoV2 suppress host type I interferon (IFN) antiviral defenses via different strategies, which is responsible for lung pathogenesis and hyper-inflammation in severe disease. NOD2 signaling following BCG vaccinations could prevent cytokine storm and hyperinflammation by different mechanisms including epigenetic reprogramming of immune cells [36]. Designing new vaccines/drugs by targeting NOD2-related molecules may provide an opportunity for controlling the hyperinflammation in SARS-Cov2 infection.

## 6. Conclusions

The role of PRRs in detecting pathogens and controlling inflammation has been well-discussed. NOD2 is a member of the NLRs in the PPR family that is involved in the detection of invading pathogens or danger signals that enter the cells. NOD2 has various interactions with other intracellular components and has a broad range of function in the induction of immune responses, in the regulation of autophagy, and in the control of ER stress. NOD2 acts cooperatively with TLRs to detect and clear invading pathogens and to produce proinflammatory cytokines. Due to NOD2′s complexity and importance, its deficiency, especially in the case of NOD2 mutations, results in exacerbated inflammation and several immune diseases. Our knowledge about the role of NOD2 in a broad range of immune reactions inside the body is now expanding. However, improving our understanding of NOD2 genetic variants and the molecular mechanisms underlying NOD2 mutation-associated diseases opens a new window for novel therapy methods based on personalized medicine, which would help to improve some conditions that are resistant to therapy or to reduce the burden of prolonged chronic diseases based on the genetic and physiological background of individual patients.

## Figures and Tables

**Figure 2 cells-10-02031-f002:**
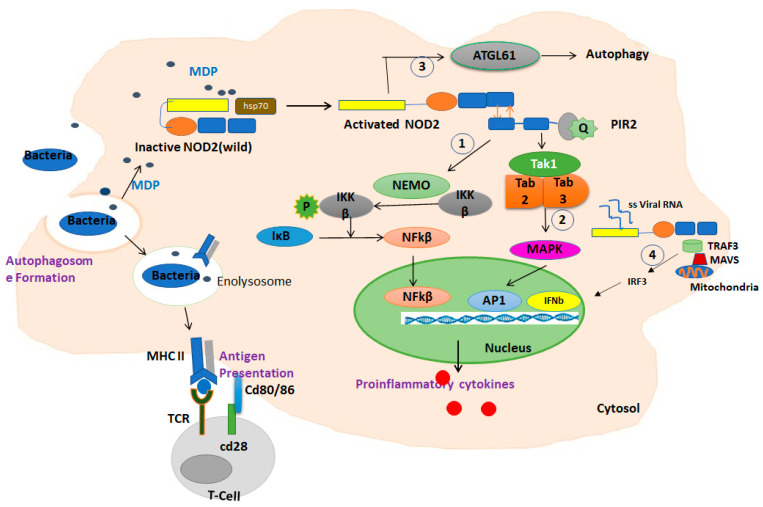
Summary of the function of NOD2 in response to MDP in macrophages. In the absence of a ligand, NOD2 is in an inactive autoinhibited form by folding the LRR domain onto NBD and CARD domains stabilized by chaperone proteins such as HSP70. Upon ligand binding and receptor activation, the C-terminal LRR domain of NOD2 undergoes a conformational change and exposes the CARD domain to interaction and oligomerization with the CARD domain in the adaptor molecule RIP2 through a homophilic CARD–CARD interaction. Upon oligomerization, PIR2 activates and promotes the ubiquitination of lysine 209 located at the kinase domain. (1) This ubiquitination promotes the recruitment of TAK1 and NEMO (the NF-kB essential modulators). The activation of TAK1 and NEMO promotes the phosphorylation of IKKβ, which is a key kinase in the NF-κB signaling pathway. Phosphorylated IKKβ degrades IκB and subsequently activates NF-κB family transcription factors. (2) In addition, this path also activates the MAPKs and AP1 pathways. The NF-κB and MAPK pathways are responsible for triggering the expression of inflammatory cytokines. (3) Activated NOD2 also may trigger an autophagic pathway by recruiting ATG16L1. (4) The interaction of NOD2/TRAF3 with mitochondrial antiviral signaling protein (MAVS) upon sensing viral ssRNA induces the activation of IRF3, triggering the expression of IFN-β gene.

**Figure 3 cells-10-02031-f003:**
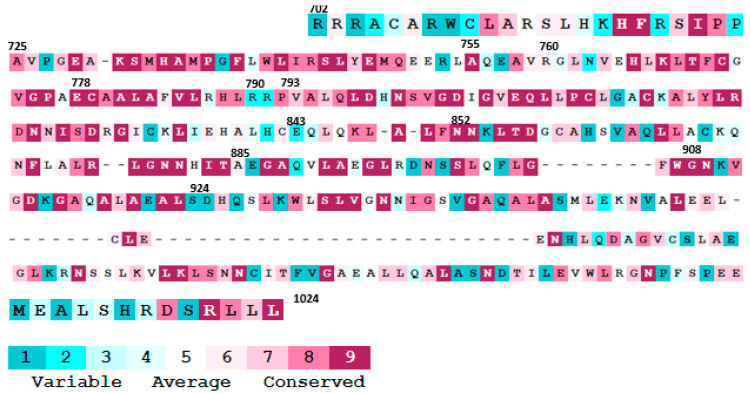
Evolutionary conservancy of the NOD2 LRR domain in the mutated positions. The image was produced by Consurf [87].

**Table 1 cells-10-02031-t001:** Some of the previous studies regarding the common polymorphisms in NOD2 gene and the associated diseases.

Number	SNPs	Mutation	Location	Population	Result	Infection (Disease)	Method	Ref
1	P268S	CCC > TCC	NBD domain	African Americans	Minor allele T is associated with a decreased risk of TB (Protective)	Tuberculosis	Sequencing of the coding regions of the NOD2 gene	[70]
	R702W	CGG > TGG [14]4	HD2 Exon 4		Minor allele T is associated with a decreased risk of TB(Protective)			
	A725G	GCT > GGT	HD2 Exon 4		the minor allele G increased the risk of TB			
2	R702W	CGG > TGG		South African	No association	Inflammatory bowel disease (CD & UC)	PCR of the Exons 4, 8 and 11- HEX-SSCP &RFLP	[71]
	A725G	GCT > GGT			Increased risk of TB			
	G908R	Rs2066845			No association			
	1007fs(insC3020)	L1007P rs5743293			No association			
3	rs3135499		Promoter	Han Chinese from Jiangsu Province	T genotype protective	Tuberculosis	TaqMan-based allelic discrimination system	[72]
	rs7194886		Promoter		Increased risk for T allele carriers			
	rs8057341		Promoter					
	rs9302752		Promoter		T genotype protective			
4	insC3020	rs5743293		Sardinian population.	Significant Association (Increased the susceptibility)	CD & Mycobacterium avium subsp. paratuberculosis	PCR & sequencing	[73]
	R702W	Rs2066844						
	G908R	Rs2066845						
5	insC3020	1007fs		northern Indian states	No mutation was observed in the patients and controls	TB and leprosy	PCR-RFLP confirmed by gene sequencing	[74]
	R702W	Rs2066844						
	G908R	rs2066845						
6	R702W			South African	No association	Tuberculosis	Tag Man platform genotyping	[75]
	G908R							
	insC3020							
7	P268S	C > T rs2066842	Exon 4	Caucasian patients	No association	Sarcoidosis	Tag Man platform genotyping	[61]
	R587R	T > G rs1861759	Exon 4					
	R702W	C > T rs2066844	Exon 4					
	G908R	G > C rs2066845	Exon 8					
	insC3020	rs2066847	Exon 11					
8	P268S			Turkish population	Association with CD	Crohn’s Disease and Ulcerative Colitis	PCR-RFLP	[76]
	M863V				No mutant was found			
9	R702W	rs2066844 CGG > TGG		Meta analysis	C allele is a risk factor	sarcoidosis	Meta-analysis	[77]
	G908R	rs2066845			no associated			
	insC3020	rs2066847			no associated			
	R587R	rs1861759			no associated			
10	C-159 T	rs2569190		Meta analysis	GG is common in TB	Tuberculosis	Meta-analysis	[78]
	A-1145G	rs2569191			T allele is a risk factor in TB			
	IV	rs1861759			TG genotype is higher in TB			
		rs7194886			T allele is a risk factor of TB			
	R702W	rs2066844			CC genotype is a risk factor for TB			
	P 507 T/S	rs2066842 C > A/T			CC genotype is a risk factor for TB			
11	-159C > T	-159C > T	promoter of CD14	Chinese	Higher risk increased promoter activity/increased sNOD2	spinal TB	Seq.	[79]
12	G-1619A	rs2915863	promoter of CD14	Han Chinese	Increased susceptibly/ increased sNOD2	tuberculosis	PCR and seq	[80]
	T-1359G	rs3138078						
	A-1145G	rs2569191						
	C-159T	rs2569190						
13	C(-159)T		promoter of CD14	Han Chinese	T allele is a RF	tuberculosis	PCR-DNA sequencing	[81]
	G(-1145)A				G allele is a RF			
14	C(-159)T		promoter of CD14		increased level of serum soluble CD14	tuberculosis		[82]
15	C(-159)T		promoter of CD14	Mexico	increased Tb susceptibility/ increased level of serum soluble CD14		PCR-RFLP	[83]
16	C(-159)T		Promoter	Meta analysis	increased risk of TB		Meta-analysis	[84]
17	R426H	rs562225614 G > A	Exon 4	Case report	Early Onset Inflammatory Bowel Phenotype	IBD-Increased expression of inflammatory cytokines	Sequencing	[85]
18	N1010K	3030A > C	LRR domain Exon 12			CD	Sequencing	[86]

**Table 2 cells-10-02031-t002:** Characterization of functionality of SNPs using sequence homology tools. The SNPs in red are predicted to be damaging by at least two of the computational homology tools.

LRR Domain SNPs	Provean Score	Role	Polyphen-2 Score	Role	PANTHER Score	Role
A725G	−1.275	Neutral	0.04	benign	455	probably Damaging
A755V	−0.942	Neutral	1	probably Damaging	456	probably Damaging
R760C	−3.651	Deleterious	0.22	benign	176	probably benign
E778K	−2.579	Deleterious	0.998	probably Damaging	455	probably Damaging
R790W	−4.021	Deleterious	0.998	probably Damaging	176	probably benign
V793M	−0.804	Neutral	0.85	probably Damaging	455	probably Damaging
E843K	0.482	Neutral	0.783	probably Damaging	176	probably benign
N852S	−3.049	Deleterious	0.998	probably Damaging	455	probably Damaging
M863V	−0.07	Neutral	0	benign	176	probably benign
A885T	−1.407	Neutral	0.835	probably Damaging	455	probably Damaging
G908R	−5.822	Deleterious	1	probably Damaging	457	probably Damaging
G924D	0.149	Neutral	0.411	benign	176	probably benign

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
