# Peer review of "Inborn Errors in the LRR Domain of Nod2 and Their Potential Consequences on the Function of the Receptor"

_cells, 2021, doi:10.3390/cells10082031_

Round 1
Reviewer 1 Report
The authors present what is indicated to be a review of inborn errors in the LRR domain of NOD2. However, the submission is basically a short general review of NOD and other PRR structure and function (up to page 7) followed by an in silico analysis of SNPs in the NOD2 gene complete with a methods and results section. It is very unusual for a review article to contain original information previously subjected to peer review. If this is to be considered an original research article, the introduction should be substantially condensed. If it is to be a review article, new data should be deleted.
Most of the "new" data, including most of the figures (along with typos) have been previously published in a preprint format (Inborn errors E778K and G908R in NOD2 gene increase risk of nontuberculous mycobacterial infection: a computational study Shamila D. Alipoor, Mehdi Mirsaeidi doi: https://doi.org/10.1101/2020.12.25.424387). While publication as a preprint does not necessarily preclude peer-reviewed publication, this should at least be acknowledged.
In figure 1 and its legend, RIP2 and PIR2 appear to be used interchangeably. Also, there are misalignments in the figure.
Author Response
The authors present what is indicated to be a review of inborn errors in the LRR domain of NOD2. However, the submission is basically a short general review of NOD and other PRR structure and function (up to page 7) followed by an in silico analysis of SNPs in the NOD2 gene complete with a methods and results section.
Q1- It is very unusual for a review article to contain original information previously subjected to peer review.
If this is to be considered an original research article, the introduction should be substantially condensed. If it is to be a review article, new data should be deleted.
R1: Thank you for great comment. We restructured the manuscript and made it as a review per recommendation. So, we first summarized the recent finding in this context and then further explained the importance of the NOD2 common mutations by prediction of the effects of these mutations on the function and structure of the receptor.
Q2- Most of the "new" data, including most of the figures (along with typos) have been previously published in a preprint format (Inborn errors E778K and G908R in NOD2 gene increase risk of nontuberculous mycobacterial infection: a computational study Shamila D. Alipoor, Mehdi Mirsaeidi doi: https://doi.org/10.1101/2020.12.25.424387). While publication as a preprint does not necessarily preclude peer-reviewed publication, this should at least be acknowledged.
R2: Thanks for comment. In the new version, we added the reference to the manuscript.
Q3- In figure 1 and its legend, RIP2 and PIR2 appear to be used interchangeably. Also, there are misalignments in the figure.
R2: Thank you for the comments. These items are corrected in the revised manuscript.
Reviewer 2 Report
Alipoor and Mirsaeidi, summarized the cellular pathways involved in the intracellular NOD-2 signaling downstream of bacterial and viral ligands. Additionally, authors performed some computational analysis to assess how NOD2 polymorphisms could affect the receptor structure and function.
Overall, the review started with basic information about PRRs, NLRs with focus on NOD2 and its association in pulmonary and genetic diseases. Later, the review turned out to be a research article and hypothesis being presented with some information in bits and pieces which needs further information and require an independent in-depth review. I feel this review needs some reorganization and I have some suggestions to make this review more interesting and appealing to a broader audience.
Major comments:
- References 1-4 are not appropriate. Ref# 25-26 cited on line 134 is not appropriate.
- Figure 1 with the domain structure is not providing any new information and is redundant with the literature. Provide some novelty to the figures by incorporating recent/new findings. For ex., add the SNPs associated with the domains with exact location. Disease associated with those mutations can also be depicted. Additionally, comparison between mouse, rat and human NOD2 domains can be drawn with percentage similarity.
- Figure 2 is not readable and should be resubmitted if it provides new information and is absolutely required.
- Most of the sections are written as multiple paragraphs of 1 or 2 sentences each (Section 1.5 on Page 6). All these sentences lack coherence and should be rewritten. Provide authors take/perspective on the previously published studies and what inference must be drawn to aid future research directions w.r.t. NOD2.
- Supplementary Table 1 should be included in the main article and needs some update with latest information (pre-clinical/clinical). Some of these polymorphisms can be described in detail, focusing on some of the well-characterized SNPs in bacterial and autoinflammatory diseases.
Minor comments:
- Some of the text in figure 1 is not legible.
- Provide appropriate reference for sentence on line 47-49.
- Ref 36 does not talk about MERS-COV viral infection.
Suggested readings:
PMID: 32677123; PMID: 33503814; PMID: 33935757; PMID: 28060562
Author Response
Reviewer 2
Alipoor and Mirsaeidi, summarized the cellular pathways involved in the intracellular NOD-2 signaling downstream of bacterial and viral ligands. Additionally, authors performed some computational analysis to assess how NOD2 polymorphisms could affect the receptor structure and function.
Overall, the review started with basic information about PRRs, NLRs with focus on NOD2 and its association in pulmonary and genetic diseases. Later, the review turned out to be a research article and hypothesis being presented with some information in bits and pieces which needs further information and require an independent in-depth review. I feel this review needs some reorganization and I have some suggestions to make this review more interesting and appealing to a broader audience.
Major comments:
Q1: References 1-4 are not appropriate. Ref# 25-26 cited on line 134 is not appropriate.
R1: Thank you for the comment. We have corrected these ref. and highlighted them in the highlighted version.
Q2: Figure 1 with the domain structure is not providing any new information and is redundant with the literature. Provide some novelty to the figures by incorporating recent/new findings. For ex., add the SNPs associated with the domains with exact location. Disease associated with those mutations can also be depicted. Additionally, comparison between mouse, rat and human NOD2 domains can be drawn with percentage similarity.
R2: The figure is revised to address the reviewer’s comments.
Q3: Figure 2 is not readable and should be resubmitted if it provides new information and is absolutely required.
R3: Thank you for mentioning this item. The figure is updated and improved for the resolution.
Q4: Most of the sections are written as multiple paragraphs of 1 or 2 sentences each (Section 1.5 on Page 6). All these sentences lack coherence and should be rewritten.
R: Thank you for the mentioning this point. We overwrite these sentences and highlighted them in the text
Q5: Provide authors take/perspective on the previously published studies and what inference must be drawn to aid future research directions w.r.t. NOD2.
R: Thank you for comment. We have added a section to the manuscript for discussing the future research directions and therapeutic opportunities regarding NOD2 context.
“….. Nod2 : Recent progress and future research perspectives
Current findings on NOD2 indicated that these molecule plays a critical role in the innate immunity and regulation of inflammatory pathways.
Stimulation of NOD2 by MDP triggers several immune pathways so this molecule has been considered as a potential therapeutic target. One of the suitable therapeutic strategies could be through its downregulation[1].
Actually, using the direct NOD2 inhibitors have not been introduced to date because of the technical challenges of targeting the oligomeric proteins. However (RIPK2) is an a key kinase in NOD2 signaling pathways [2] [3]
The RIPK2 inhibitors inhibited the murine experimental colitis through the suppression of the polyubiquitination and kinase activity of RIPK2[4,5]. Another study suggested that the ATP-binding domain in RIPK2 also could be targeted to interfere with the RIPK2-XIAP interaction and modulation of NOD signaling[6]. Recent experimental and clinical studies have focused on the possibility of using RIPK2 inhibitors as a new therapeutic target in NOD2 dependent auto-immune diseases including IBD[2,7].
Targeting of the NOD2 immune pathways also have been interested as vaccine adjuvants. [4].
Using TLR4 ligand (MPLA) as vaccine adjuvant to stimulates immunityagainst influenza antigens was approved for the first time in 2017. However; Using PRR agonists such as NLRP3[8], TLR3[5], TLR7[6]orTLR9[7]ligands as vaccine adjuvants are currently examined in clinical trials,.
NOD2 agonists are interested as the efficient mucosal adjuvants. Recently a new chimeric TLR7/NOD2 ligand is introduced as a potent adjuvant to stimulate both systemic and mucosal immune responses through an optimal DC maturation process [9].
Furthermore; MIS416 ; a non-toxic microparticle . is recently introduced as cancer vaccine adjutant in a first-in-human clinical trial and activates the immune functions via NOD2 and TLR9 related pathways[10].
In addition it is suggested that NOD2 may play an important role in the outcome of kidney cancer patients AND The the expression level of NOD2 gene is associated with the prognosis of kidney cancer patients and has potential as a biomarker for the survival of kidney cancer patients.
NOD2 may also play a role in carcinogenesis via regulation apoptosis and inflammation. NOD2 variants have been reported to be associated with risk of cancers in Caucasian population[11-13] . So mediating of NOD2 signaling pathways molecules could be a target of cancer therapy methods.
Interestingly it is recently suggested that BCG-induced trained immunity via NOD2 may play a key role for the control of SARS-CoV2 infection. SARS-CoV2 suppress host type I interferon (IFN) antiviral defenses via different strategies which is responsible for lung pathogenesis and hyper-inflammation in severe disease. It is suggested that NOD2 signaling following BCG vaccinations could prevent cytokine storm and hyper inflammation by different mechanisms including epigenetic reprogramming of immune cells [14]. So this indicate NOD2 targeting and designing new vaccine by targeting NOD2 related molecules may provide an opportunity of /controlling the hyperinflammation in SARS-Cov2 infection.”
Q5: Supplementary Table 1 should be included in the main article and needs some update with latest information (pre-clinical/clinical). Some of these polymorphisms can be described in detail, focusing on some of the well-characterized SNPs in bacterial and autoinflammatory diseases.
R5: The table is updated and included in the main text
Minor comments:
Q1: Some of the text in figure 1 is not legible.
R1: Thank you for comment. These items are corrected in figure 1
Q2: Provide appropriate reference for sentence on line 47-49.
R2: The reference has been provided:
Trindade, B.C.; Chen, G.Y. NOD1 and NOD2 in inflammatory and infectious diseases. Immunological Reviews 2020, 297, 139-161.
Reviewer 3 Report
I have read with great interest the study by Alipour an Mirsaedi, focused on the errors in the LRR domain of NOD2 and their consequences in the receptor's function. This is a very well written study and carefully structured study which reveal the latest information on the corresponding field. The figures read very nice and the layout is very attractive to the audience, the tables appear in the right context and the conclusions are complete and provide the authors' insight in their field of endeavour. The references are complete, relevant to the study and up to date Hence, I highly reccommend the publication of the manuscript. My only comment is that the title should be changed, so to read as......."on the function of the receptor..."
Author Response
Reviewer 3:
I have read with great interest the study by Alipour an Mirsaedi, focused on the errors in the LRR domain of NOD2 and their consequences in the receptor's function. This is a very well written study and carefully structured study which reveal the latest information on the corresponding field. The figures read very nice and the layout is very attractive to the audience, the tables appear in the right context and the conclusions are complete and provide the authors' insight in their field of endeavour. The references are complete, relevant to the study and up to date Hence, I highly reccommend the publication of the manuscript. My only comment is that the title should be changed, so to read as......."on the function of the receptor..."
R: Thank you for the comments. We revised the title of manuscript to address your concern.
Round 2
Reviewer 1 Report
The manuscript has been extensively revised to focus on a review of NOD2 function. It is informative and should be a useful addition to the literature.
Reviewer 2 Report
The review article has been substantially improved and all of my comments have been addressed.